# Enhanced *Agrobacterium*-Mediated Transformation in Chinese Cabbage via Tenoxicam, Phytohormone Optimization, and Visual Reporters

**DOI:** 10.3390/plants14243802

**Published:** 2025-12-13

**Authors:** Shubin Wang, Shuantao Liu, Ronghua Wang, Zhigang Zhang, Nianfang Xu, Qiaoyun Li, Zhizhong Zhao

**Affiliations:** Shandong Key Laboratory of Bulk Open-Field Vegetable Breeding, Institute of Vegetables, Shandong Academy of Agricultural Sciences, Jinan 250100, China; liushuantao870@163.com (S.L.); wangrh1101@163.com (R.W.); zhangzhigang7781@163.com (Z.Z.); xunianfang2009@163.com (N.X.); liqiaoyun0606@126.com (Q.L.)

**Keywords:** Chinese cabbage, *Agrobacterium*-mediated transformation, tenoxicam, eYGFPuv, RUBY

## Abstract

Chinese cabbage (*Brassica rapa* ssp. *pekinensis*) is a globally important leafy vegetable, but functional genomics research on its recalcitrance to *Agrobacterium*-mediated genetic transformation is severely limited. In this study, we demonstrate that both *Agrobacterium* infection and antibiotic selection significantly inhibit cotyledonary petiole regeneration, representing one principal bottleneck to high-throughput transformation. Infection with different *Agrobacterium* strains suppressed the regenerated shoot per explant by 30.98–69.16%. Supplying the salicylic acid signaling inhibitor tenoxicam in the seed germination medium raised post-infection regeneration by up to 37.90%. Compared with non-infected controls, the optimal NAA concentration for explant regeneration after infection was higher, and 0.5 mg/L increased post-infection regeneration by 27.66%. Replacing antibiotic selectable markers with the visual reporter eYGFPuv or RUBY eliminated phytotoxicity, reduced false-positive shoots, and further elevated transformation efficiency to 19.33–20.00% (versus 2.67–6.67% under antibiotic selection). The integrated protocol yielded stable RUBY overexpressing lines, the biomass of which declined with rising transcript levels. Restricting RUBY expression to the inner head leaves generated a novel germplasm with less yield penalty. This work provides a high-efficiency transformation method that will accelerate gene discovery and genome editing in Chinese cabbage.

## 1. Introduction

Chinese cabbage (*Brassica rapa* ssp. *pekinensis*) is the most consumed leafy vegetable in Asian countries [1]. China has the world’s largest cultivation area for Chinese cabbage, with an annual area of 1.73–1.87 million hectares [2]. Since the release of the reference genome [3], the identification and functional characterization of genes underlying key agronomic traits have become the central objective of functional genomics in Chinese cabbage [4,5]. *Agrobacterium*-mediated transformation is an essential tool for gene function analysis, genome editing, and genetic engineering in plants [6,7,8,9]. In Chinese cabbage, *Agrobacterium*-mediated transformation and/or genome editing have been used to functionally characterize genes underlying self-incompatibility [10], agronomic traits [11,12,13], biotic [14,15] and abiotic stress [16,17] resistance, and nutrient enhancement [18].

However, Chinese cabbage is considered the most recalcitrant *Brassica* species for *Agrobacterium*-mediated transformation [19,20]. Over the past three decades, genotype screening, medium optimization and protocol refinement have incrementally raised, though only modestly, the transformation efficiency in certain Chinese cabbage genotypes [21,22,23,24,25]. Nevertheless, genetic transformation in Chinese cabbage remains far less exploited than in highly transformable *Brassica* species such as *B*. *napus* and *B*. *oleracea*. In recent years, co-transformation of target genes together with developmental regulators has dramatically increased the transformation efficiency of maize, wheat, and several other crop species [26,27,28]. More recently, this strategy has been successfully extended to Chinese cabbage [29] and other *B*. *rapa* subspecies such as Bok choy and Pai-Tsai [30]. However, most developmental regulators confer pleiotropic effects, such as reduced fertility, curled leaves, and other overt phenotypic changes [26,30], which restrict their deployment in specific scenarios. Achieving a truly efficient genetic transformation protocol for Chinese cabbage thus remains challenging.

In genetic transformation, *Agrobacterium* acts as the DNA delivery vector, but exerts pronounced phytotoxicity associated with plant immune signaling and endogenous hormone homeostasis [31]. It is well established that Ethylene (ET)-, salicylic acid (SA)-, and reactive oxygen species (ROS)-mediated immune responses are deployed by plants to restrict *Agrobacterium*-mediated transformation [32,33,34,35]. Manipulating plant immune responses represents an effective strategy for enhancing *Agrobacterium*-mediated genetic transformation. Pharmacological inhibitors of plant immune signaling, such as 1-aminocyclopropane-1-carboxylate (ET signaling inhibitor) [36], tenoxicam (SA signaling inhibitor) [37], LaCl_3_ (lanthanum chloride, a Ca^+2^ channel blocker) [38], and L-cysteine (a ROS inhibitor) [25], have been shown to elevate transformation efficiency in diverse plant species.

In plant genetic transformation, selectable markers are essential for eliminating non-transformed cells and selectively enriching transformed cells [39]. In Chinese cabbage, kanamycin and hygromycin are the most widely used selection reagents [25]. However, these antibiotic selection systems cannot fully eliminate false positives and, more critically, selection reagents exert a certain inhibitory effect on the growth and differentiation of transformed cells [40,41]. To circumvent these drawbacks, visual reporters such as β-glucuronidase (GUS), fluorescent proteins, and RUBY have been increasingly adopted. The visual reporters enable rapid, accurate, and non-invasive tracking of gene expression and genetic transformation [42,43,44,45], and screening transformants solely by visual reporters without any selectable reagents has been reported to reduce false-positive rates, increase transformation efficiency, and shorten the overall workflow [46,47].

In this study, we demonstrate that both *Agrobacterium* infection and selectable reagent markedly suppress cotyledonary petiole regeneration in Chinese cabbage, constituting a major bottleneck to transformation efficiency. Supplying tenoxicam during seed germination and elevating 1-naphthaleneacetic acid (NAA) in the shoot induction medium significantly improve the regeneration efficiency of explants after *Agrobacterium* infection. The visual reporters eYGFPuv [43] and RUBY [44] could fully replace antibiotic selectable markers for positive shoot selection and enhance explant transformation frequency. Integrating these findings, we established a more efficient genetic transformation protocol for Chinese cabbage.

## 2. Materials and Methods

### 2.1. Plant Materials

A total of 110 Chinese cabbage accessions were employed as experimental materials, comprising 29 commercial cultivars purchased from the retail market and 71 inbred lines developed by our research group (Appendix A). All seed lots are maintained at the Institute of Vegetables, Shandong Academy of Agricultural Sciences, Jinan, China.

### 2.2. Vector Construction and Agrobacterium Strain

The UV-excitable green fluorescent protein (GPF) eYGFPuv [43] and naked-eye-visible betalain-producing reporter gene RUBY [44] were exploited to roughly quantify *Agrobacterium*-mediated transient expression efficiency and to enable non-destructive in planta screening of transgenic adventitious shoots. Vectors pCAMBIA2300 and pCAMBIA1300 [48], which confer kanamycin and hygromycin resistance, respectively, serve as backbone vectors. The eYGFPuv and RUBY expression cassettes, each composed of the double-enhanced CaMV 35S (2X35S) promoter [49], 5′-UTR from *Arabidopsis thaliana* gene *cold*-*regulated 47* (*COR47*) [50], the eYGFPuv [43] or RUBY [44] coding sequence, and the nopaline synthase (NOS) terminator [51], were commercially synthesized as intact fragments by BGI Genomics and subsequently inserted between the EcoRI and PstI sites of the vectors pCAMBIA2300 and pCAMBIA1300. The resulting vectors were introduced into *A*. *tumefaciens* strains GV3101 and LBA4404, or *A*. *rhizogenes* strain K599 via freeze–thaw transformation.

### 2.3. Culture Media

Murashige and Skoog (MS) medium [52] containing 30 g/L sucrose and 9 g/L agar (pH 5.8) served as the basal medium. Co-cultivation medium consisted of the basal MS supplemented with 4 mg/L 6-benzylaminopurine (6-BA), 0.2 mg/L NAA, and 200 μM acetosyringone. Shoot induction medium was the basal MS containing 4 mg/L 6-BA, 0.2 mg/L NAA, 6 mg/L AgNO_3_, and 400 mg/L timentin.

### 2.4. Cotyledonary Petiole Tissue Culture

Uniform and intact seeds were selected, and any seeds with cracked seed coats were discarded. After a brief rinse in sterile water, seeds were surface-sterilized in 75% ethanol for 1 min followed by 2% NaClO for 10 min, then washed twice with sterile water and blotted dry. Thirty seeds were evenly sown on seed germination medium (i.e., the basal MS medium) in each Petri dish (90 mm diameter, 30 mm height). Dishes were sealed and incubated at 25 °C under a 14 h light/10 h dark photoperiod. After 4–5 d, cotyledons were aseptically excised and inserted vertically (petiole end down) into shoot induction medium. Cultures were returned to the same growth condition as seed germination.

### 2.5. Genetic Transformation

A glycerol stock of *Agrobacterium* stored at −80 °C was inoculated into LB medium containing 25 mg/L of each antibiotic (rifampicin and kanamycin for GV3101 and LBA4404; streptomycin and kanamycin for K599) and incubated at 28 °C with 150 rpm shaking for 24 h. For infection solution preparation, approximately 100 μL of the overnight culture was added to 20 mL LB medium, and the OD600 was then adjusted to 0.2. After excision, cotyledonary petioles were briefly dipped in the infection solution for 2 s and inserted into co-cultivation medium, with 40–50 explants distributed evenly per plate. Explants were kept in darkness at 23 °C for 3 d, then transferred to shoot-induction medium (10 petioles per plate) and cultured at 25 °C under a 14 h light/10 h dark. When antibiotic selection was required, explants were first cultured on antibiotic-free shoot induction medium for 5 d, and then transferred to the same medium supplemented with the appropriate antibiotic. The concentrations of kanamycin and hygromycin used for positive shoot screening were 25 mg/L and 10 mg/L, respectively [23,24].

### 2.6. Assessment of Regeneration and Transformation Efficiency

The visual reporter genes eYGFPuv or RUBY were used to discriminate between transgenic and non-transgenic shoots. eYGFPuv fluorescence was checked using a 365 nm UV flashlight [43]. *Agrobacterium* infection and/or antibiotic selection delayed shoot regeneration compared with non-infected, non-antibiotic-treated controls. After 25–30 d on shoot induction medium, all explants including non-inoculated, inoculated without selection, or inoculated plus antibiotic selection, had ceased producing new shoots, and total, positive, and negative shoot numbers were recorded. For each treatment, three replicates were performed, each comprising 50 explants. Regeneration efficiency was defined as the mean number of regenerated shoots per explants. Transformation efficiency was defined as the number of positive shoots divided by the total number of explants.

### 2.7. Agronomic Trait Evaluation of Transgenic Lines

Positive T0 plantlets were vernalized at 4 °C for 20 d in a growth chamber and transplanted to the greenhouse. T1 and T2 transgenic seeds were germinated in plug trays, and 15-day-old seedlings transplanted to the field at 40 cm spacing. For agronomic trait evaluation, a randomized complete block design was adopted with three replications for transgenic lines. Each block contained 20 plants, from which 5 were randomly selected for phenotypic measurement. Recorded traits included growth period, gross plant weight, net head weight, head height, head width, and total number of head leaves.

### 2.8. RNA Extraction, RT-PCR, and qRT-PCR Analysis

The total RNA was extracted using the ‘FastPure^®^ Universal Plant Total RNA Isolation Kit’ (Vazyme, Code No. RC411, Nanjing, China). The synthesis of cDNA from RNA was performed by using ‘HiScript^®^ III RT SuperMix for qPCR’ (Vazyme, Code No. R323, Nanjing, China). Specific primers CYP76AD1-F (5′-TCGCCAAGATTCACGGCC-3′) and CYP76AD1-R (5′-TATTCGGGATCGTGCGGTTG-3′) were designed to detect the expression of RUBY by quantitative real-time PCR (qRT-PCR). Actin homologs in Chinese cabbage were used as reference RNA [18]. For qRT-PCR, three biological replicates were performed for each sample. The relative gene expression was calculated using the 2^−∆CT^ method.

### 2.9. Statistical Analysis

The R software v4.1.2 was used for statistical analysis [53]. The R function ‘chisq.test’ was used for performing the Chi-square test, and the R package ‘agricolae’ was used to perform the Least Significant Difference (LSD) multiple-comparison test. The significance level of the LSD test is set to *p* < 0.05.

## 3. Results

### 3.1. Variation in Regeneration Efficiency of Cotyledonary Petioles Among Different Genotypes

First, cotyledonary petioles of 110 Chinese cabbage accessions were evaluated for in vitro shoot regeneration. After 25–30 days of culture on shoot induction medium, the number of adventitious shoots induced from explants was assessed (Figure 1; Appendix A). The materials were categorized into three distinct types based on the regeneration response: Type I (*n =* 71, 64.5%) produced compact callus followed by differentiation of both adventitious shoots and roots; Type II (*n* = 29, 26.4%) formed callus but failed to initiate organogenesis; Type III (*n* = 10, 9.1%) exhibited rapid browning and necrosis at the cut surface without callus formation. Within the regenerable Type I accessions, regeneration efficiency varied extensively across genotypes (Figure 1d). Accessions with high regeneration efficiency (≥10 shoots per explant) numbered 14, accounting for the smallest proportion (12.7%); accessions with moderate (5–10 shoots per explant) and low (<5 shoots per explant) regeneration efficiency numbered 28 (25.5%) and 68 (61.8%), respectively.

### 3.2. Impact of Agrobacterium Infection on the Regeneration Capacity of Cotyledonary Petioles

To investigate the impact of *Agrobacterium* infection on explant regeneration, cotyledonary petioles of the highly regenerable accessions L68, L69, and L131 were separately inoculated with the *A. tumefaciens* strains GV3101 and LBA4404, and an *A*. *rhizogenes* strain K599, each carrying the binary vector pCAMBIA2300-RUBY. No selectable reagent was added to the shoot induction medium. After *Agrobacterium* infection, the regeneration efficiency of cotyledonary petioles declined markedly in all genotypes, albeit with only minor differences among genotypes (Table 1, Figure 2, Appendix A). All *Agrobacterium* strains significantly inhibited explant regeneration. GV3101 and LBA4404 were the most suppressive, reducing regeneration by 54.44–59.85% and 63.69–69.16%, respectively, whereas K599 caused a smaller decrease (30.98–50.17%). Accession L68 and *Agrobacterium* strain GV3101 were used for subsequent experiments.

### 3.3. Screening of Chemical Reagents That Promote Cotyledonary Petiole Regeneration After Agrobacterium Infection

The effects of three chemical inhibitors of plant immune signaling on the regeneration efficiency of cotyledonary petioles after *Agrobacterium* infection were evaluated using accession L68 (Appendix A). These chemicals include the SA pathway inhibitors tenoxicam [37] and CNTQ [54], and the Ca^+2^ channel inhibitor LaCl_3_ (lanthanum chloride) [38], all of which have previously been reported to enhance either transformation efficiency or explant regeneration in plants. We also tested trichostatin A, an epigenetic inhibitor previously shown to increase microspore regeneration in Ornamental Kale [55] and *Brassica napus* [56]. The reagents were supplied separately in seed germination, co-cultivation, or shoot induction media. *Agrobacterium* strain GV3101 carrying pCAMBIA2300-RUBY was used for infection.

Across all reagent-stage combinations, only tenoxicam applied during seed germination markedly enhanced post-infection regeneration, increasing shoot number per explant by 37.90% relative to the control (Table 2). Adding tenoxicam during co-culture or shoot induction had no significant effect on explant regeneration. CNQX and trichostatin A had no significant effect on regeneration at any of the tissue culture stages tested. LaCl_3_ supplementation during germination or co-cultivation had no significant effect, whereas its addition at the differentiation stage markedly inhibited explant regeneration.

In addition, after infection with *Agrobacterium* strain GV3101 carrying pCAMBIA2300-RUBY, explants from the tenoxicam seed germination treatment showed markedly stronger RUBY expression (larger red area) at the cut surface after co-culture and subsequent shoot induction stages, indicating that tenoxicam also enhances *Agrobacterium* infection efficiency (Figure 3).

### 3.4. Effects of 6-BA and NAA Concentrations on the Regeneration Efficiency of Cotyledonary Petioles After Agrobacterium Infection

In plant regeneration, phytohormones are critical for explant differentiation, and during transformation, *Agrobacterium* infection and wounding can disrupt their balance [31,54,57]. We compared the regeneration capacity of explants with and without *Agrobacterium* infection under various concentrations of 6-BA and NAA using accession L68 (Appendix A). *Agrobacterium* strain GV3101 carrying pCAMBIA2300 was used for infection.

The combination of 4 mg/L 6-BA and 0.2 mg/L NAA used during high regenerable accession screening was defined as the standard. In the absence of *Agrobacterium* infection, this hormone ratio proved optimal for adventitious shoot induction (Table 3). After *Agrobacterium* infection, raising the NAA concentration to 0.5 mg/L maximized adventitious shoot production at 7.43 shoots per explant, a 27.66% increase over the 0.2 mg/L level. After infection, changing the 6-BA concentration produced a similar extent of decline on regeneration efficiency to that seen without infection.

### 3.5. Comparison of Selectable Markers and Visual Reporters for Transformation Efficiency

Based on these results, we optimized the medium formula: 50 μM tenoxicam was added to the germination medium, and the NAA concentration in both co-cultivation and shoot-induction medium was raised from 0.2 mg/L to 0.5 mg/L. Using this optimized medium, we compared the performance of the antibiotic selectable markers kanamycin and hygromycin with that of a visible reporter genes eYGFPuv and RUBY in genetic transformation (Table 4, Appendix A).

In regenerated shoots, both eYGFPuv and RUBY are highly discernible, facilitating unambiguous discrimination between transgenic and non-transgenic shoots and the subsequent production of healthy transgenic seedlings (Figure 4). Kanamycin and hygromycin effectively inhibited the formation of non-transgenic seedlings, but still yielded 82.81% and 84.72% false-positive shoots, respectively. Using eYGFPuv and RUBY without selectable reagents allowed numerous non-transgenic shoots to develop; nevertheless, the number of positive shoots rose markedly compared with kanamycin or hygromycin selection, ultimately yielding a higher overall transformation efficiency (19.33% and 20.00%, respectively).

### 3.6. Genetic Segregation and Phenotypic Characterization of RUBY Overexpression Plants

Currently, RUBY is widely used as an indicator marker for genetic manipulation [29,44,45,58] and germplasm innovation of colored crops [59,60,61]. In this study, three representative T0 plants (RB16, RB25, and RB52) transformed with pCAMBIA2300-RUBY, together with their T1 and T2 progeny, were analyzed to determine the segregation of the transgene, its genetic stability, and the impact of RUBY expression on Chinese cabbage growth (Appendix A).

The T0 seedlings derived from tissue cultures were vernalized at 4 °C for 30 days and subsequently transplanted to the field, and therefore, they failed to form heads and directly bolting and flowering. All T0 plants exhibiting complete fertility and RUBY was significantly expressed in most tissues, including leaves, floral stalks, petals, and anthers, though with relatively low expression levels in floral stalks (Figure 5a,b). In the T1 generation, RUBY was clearly expressed in the cotyledons (Figure 5c). RB16 and RB25 produced red–green segregation ratios of 101:42 and 110:32, respectively, both fitting a single-locus insertion (3:1; χ^2^ = 1.45, *p* = 0.23 and χ^2^ = 0.46, *p* = 0.50, respectively), while the segregation ratio of red to green plants in RB52 was 131:13, consistent with the double-copy insertion pattern (15:1; χ^2^ = 1.90, *p* = 0.17).

According to the T2 homologous lines, significant differences in RUBY expression patterns were observed among the three representative lines: the double-copy line RB52 showed high RUBY expression in all tissues (Figure 5j–l); the single-copy line RB16 had detectable RUBY expression in all tissues except the leaf midribs (Figure 5d–f); and the single-copy line RB25 exhibited high RUBY expression only in the inner head leaves (Figure 5g–i). We further used qRT-PCR to detect RUBY expression levels in true leaves at the seedling stage, outer leaves, and middle head leaves at the heading stage (Appendix A). RB52 and RB16 showed similar expression patterns in all tested tissues, but the expression level of RB52 was significantly higher than that of RB16; in RB25, RUBY had relatively high expression in the inner head leaves but low expression in true leaves at the seedling stage and outer leaves at the heading stage, and these results were completely consistent with visual observations.

Agronomic traits were measured using the T2 transgenic lines (Table 5, Appendix A). RUBY expression significantly reduced the net weight. Moreover, among RB25, RB16, and RB52, the inhibitory effect was gradually enhanced with the increase in RUBY expression level, and lines with relatively smaller head weight had a shorter growth period, while RUBY expression had no significant effect on the number of outer leaves or head leaves.

## 4. Discussion

Among *Brassica* crops, Chinese cabbage is the most recalcitrant to transformation, chiefly because of its poor explant embryogenesis efficiency in vitro. A survey of 123 accessions identified only 17 (13.8%) with >80% cotyledonary petiole regeneration frequency, and just 1 of these yielded >10 shoots per explant [62]. In this study, screening of 110 accessions uncovered a broader set of highly regenerable genotypes, with 14 lines (12.73%) producing >10 adventitious shoots per explant (Figure 1d). These findings indicate that Chinese cabbage cotyledonary petiole regeneration is highly genotype-dependent, and only a small fraction of accessions possess strong regenerative capacity.

Reports indicate that *Agrobacterium* infection activates immune signaling, triggers ROS bursts, and disrupts hormone balance in plants [31]. Some of these alterations lead plants to resist *Agrobacterium*, impair callus formation and differentiation, and ultimately reduce transformation efficiency [32,33,34,35]. We found that the regeneration capacity of Chinese cabbage cotyledonary petioles is highly sensitive to *Agrobacterium* infection. In three highly regenerable genotypes, the number of regenerated shoots per explant declined by 30.98–69.16% after inoculation with different *Agrobacterium* strains (Table 1), indicating that *Agrobacterium* suppression of regeneration is a major barrier to efficient transformation of Chinese cabbage.

The discovery of plant immune chemical inhibitors offers a new avenue for enhancing plant transformation efficiency [36,37,38]. For example, AgNO_3_, an inhibitor of the ET signaling pathway, is routinely included in the majority of *Agrobacterium*-mediated plant transformation protocols [63]. In this study, we screened several previously reported immune inhibitors and found that tenoxicam significantly enhanced both *Agrobacterium* infection efficiency and post-infection regeneration (Table 2, Figure 2). Tenoxicam has previously been shown to enhance *Agrobacterium*-mediated transient transformation of *Arabidopsis* leaves, genetic transformation of Jatropha curcas cotyledons, and survival of maize immature embryo calli, but it exerts no significant effect on transformation in *Brassica rapa*, *Brassica napus*, oilseed rape, rice, or soybean [37]. In this study, the observation that tenoxicam enhances transformation only when applied during seed germination, while having no effect during co-cultivation and differentiation (Table 2), highlights the critical importance of timing in chemical enhancement of transformation efficiency. Among the chemicals we tested, LaCl_3_ had previously been reported to elevate maize transformation efficiency [38], and Trichostatin A to enhance microspore regeneration in Ornamental Kale [55] and Brassica napus [56]. However, neither compound improved Chinese cabbage cotyledonary-petiole regeneration (Table 2). Additionally, we found that the optimal NAA concentration for maximal regeneration differs between non-infected explants and those exposed to *Agrobacterium* (Table 3). This finding has not been reported previously.

Kanamycin and hygromycin are the selectable markers most commonly employed for Chinese cabbage transformation [25]. Multiple studies have shown that both kanamycin and hygromycin can inhibit the regeneration of transformed cells to some extent in different plant species [40,41]. Visual reporter genes enable accurate identification of transformed plants, but they confer no selective advantage to transformed cells and are therefore routinely used alongside selectable markers. However, several reports have employed visual markers as the sole means of screening and achieved improved transformation efficiency [46,47]. In this study, using eYGFPuv or RUBY in place of antibiotics for screening positive shoots yielded higher transformation efficiency while proving more accurate and convenient (Table 4).

Owing to the red color and health-promoting properties of betalains, RUBY has attracted commercial interest and has been applied to the creation of novel germplasm in crops such as rice, carrot, and cotton [59,60,61]. In this study, we generated homozygous RUBY-overexpressing lines of Chinese cabbage, creating novel red-pigmented germplasm. Since the RUBY’s substrate tyrosine is a core metabolic resource, previous studies have suggested that high-level RUBY expression could sequester tyrosine and thereby compromise plant growth and development [60,61]. We selected three stably expressing lines that differed significantly in transcript level and evaluated their agronomic traits. RUBY expression markedly reduced plant size and biomass without altering external proportions, and the degree of inhibition increased with RUBY transcript abundance (Table 5). Previous report in carrot showed that tailoring the RUBY expression pattern is an effective way to mitigate the pleiotropic effects of RUBY overexpression [62]. Notably, we obtained a stably inherited line (RB25) that shows high expression in the inner head leaves but low expression in the rosette and outer head leaves, largely preserving yield while introducing a distinctive red-pigmented Chinese cabbage type (Figure 5g–i). This specific expression pattern may be related to positional effects caused by the insertion site of the expression cassette in the genome [64].

The transformation protocol established here relies on genotypes with inherently high regeneration capacity and is therefore unsuitable for materials that cannot regenerate or exhibit very low regeneration rates. Elucidating the genetic and molecular mechanisms underlying regeneration in Chinese cabbage is crucial for overcoming genotype-dependent limitations in genetic transformation. Several studies have reported that *A*. *rhizogenes* yields higher transformation efficiency than *A*. *tumefaciens* [58,65]. We found that *A*. *rhizogenes* strain K599 imposes a significantly weaker inhibition on explant regeneration than *A*. *tumefaciens* strain GV3101 or LBA4404. However, as *A*. rhizogenes delivers not only the target gene but also *rol* genes on helper T-DNA that can exert pleiotropic effects on plant development [65], we did not use *A*. *rhizogenes* as the infection strain.

## 5. Conclusions

In this study, we show that both *Agrobacterium* infection and antibiotic selection severely inhibit cotyledonary petiole regeneration in Chinese cabbage in a previously underappreciated manner, representing a major bottleneck to high-throughput transformation. The application of the pharmacological inhibitor tenoxicam during seed germination, together with a simple upward adjustment of NAA in the shoot induction medium, largely removes this block and enhances regeneration capacity. Replacing antibiotics with the visible reporters eYGFPuv or RUBY eliminates phytotoxicity, reduces false positives, and further elevates transformation frequency. Combined, these three refinements yield a highly efficient protocol for genetic transformation in Chinese cabbage.

## Figures and Tables

**Figure 1 plants-14-03802-f001:**
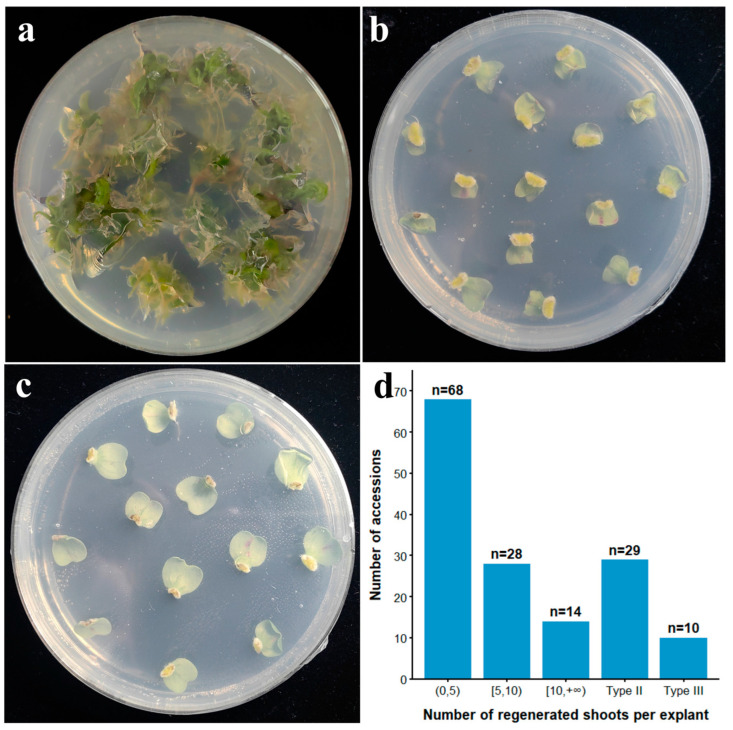
Three distinct regeneration responses of cotyledonary petioles. (**a**) Type I, normal regeneration (accession L68); (**b**) Type II, callus forming but non-regenerating (accession L56); (**c**) Type III, cut surface browning without callus formation and regeneration (accession 39); (**d**) distribution of accessions across different regeneration efficiency.

**Figure 2 plants-14-03802-f002:**
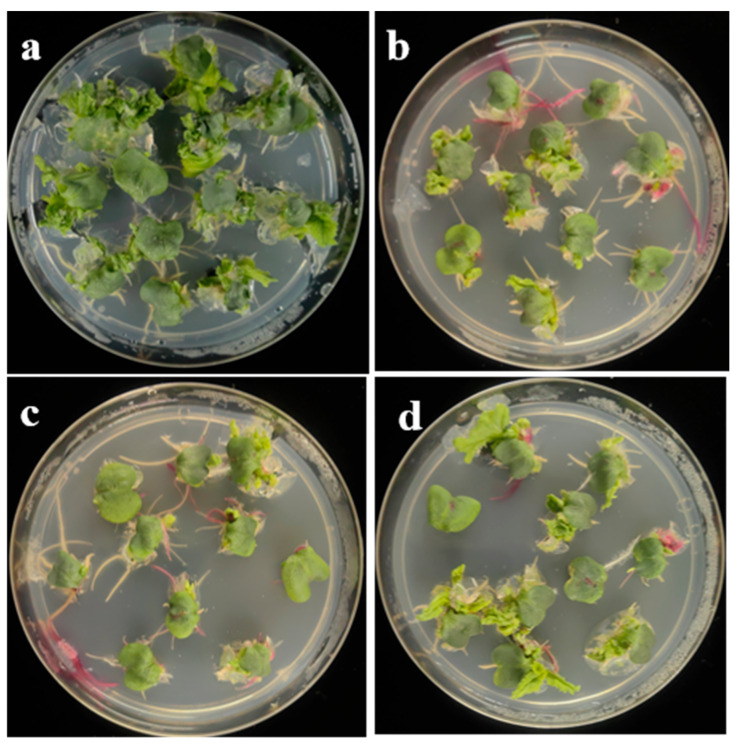
Differences in adventitious shoot regeneration of accession L68 with GV3101, LBA4404, and K599 infection, as compared to non-infected control. (**a**) Non-infected controls; (**b**–**d**) infected with GV3101, LBA4404, and K599, respectively. Red tissues indicate RUBY expression.

**Figure 3 plants-14-03802-f003:**
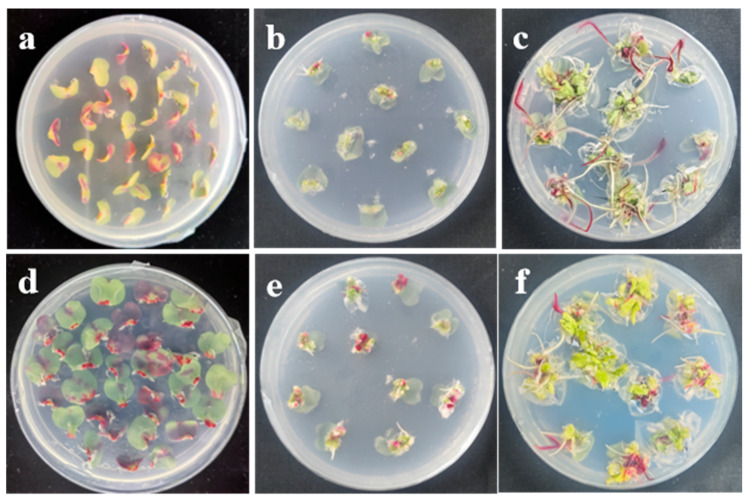
RUBY expression at different stages in explant without tenoxicam treatment (CK; (**a**–**c**)) explants without tenoxicam treatment on day 3 of co-cultivation medium, day 10 of shoot induction medium, and day 30 of shoot induction medium; (**d**–**f**) explants treated with tenoxicam on day 3 of co-cultivation medium, day 10 of shoot induction medium, and day 30 of shoot induction medium.

**Figure 4 plants-14-03802-f004:**
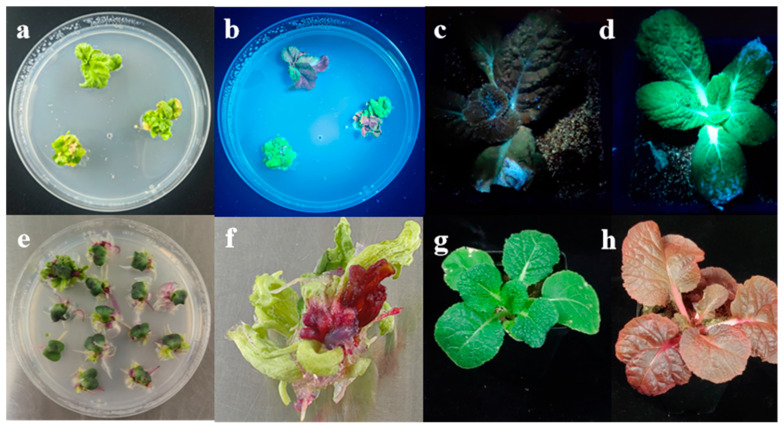
The expression of visual reporters eYGFPuv and RUBY in positive shoots and transgenic seedlings. (**a**,**b**) Expression of eYGFPuv in cultured explants under white light and UV illumination, respectively; (**c**,**d**) transgenic seedling of eYGFPuv and non-transgenic control under UV illumination, respectively; (**e**,**f**) expression of RUBY in cultured explants; (**g**) non-transgenic control; (**h**) transgenic seedlings overexpressing RUBY.

**Figure 5 plants-14-03802-f005:**
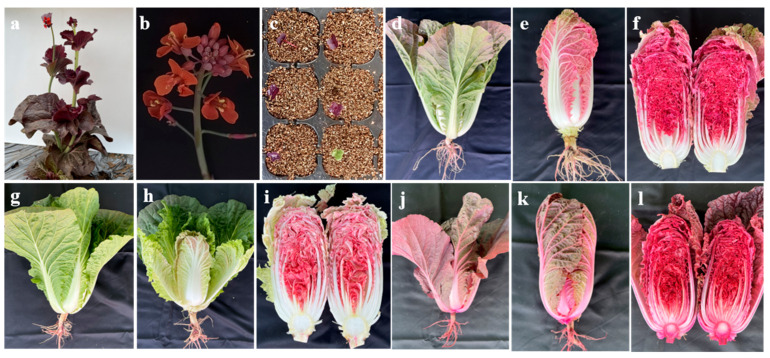
Morphological phenotype of RUBY overexpressing lines. (**a**) Bolting stage of T0 plant; (**b**) flowers of T0 plant; (**c**) cotyledons of T1 plant; (**d**–**f**) leaf head and cross-section of transgenic line RB16 at T3 generation; (**g**–**i**) leaf head and cross-section of transgenic line RB25 at T3 generation; (**j**–**l**) leaf head and cross-section of transgenic line RB52 at T3 generation.

**Table 1 plants-14-03802-t001:** Average number of regenerated shoots per explant after non-infection and infection with different *Agrobacterium* strains.

Accession	Non-Infection	GV3101	LBA4404	K599
L68	12.52 ± 0.39 ^a^	5.70 ± 0.47 ^c^	4.28 ± 0.63 ^d^	6.24 ± 0.68 ^b^
L69	10.46 ± 0.54 ^a^	4.52 ± 0.32 ^c^	5.18 ± 0.58 ^c^	7.22 ± 0.66 ^b^
L131	11.69 ± 0.64 ^a^	4.24 ± 0.64 ^c^	3.68 ± 0.66 ^c^	6.54 ± 0.67 ^b^

Values are presented as mean ± standard deviation (SD; n = 3). For each accession, different letters after SD indicate significant differences among treatments according to LSD test at the 0.05 level.

**Table 2 plants-14-03802-t002:** The average number of regenerated shoots per explant after chemical reagents application at different culture stages.

Reagent	Concentration	Treatment Stage	Regenerated Shoots per Explant
CK	/	/	6.20 ± 0.38
Tenoxicam	50 μM	Seed germination	8.55 ± 0.36 *
Tenoxicam	50 μM	Co-cultivation	6.15 ± 0.39
Tenoxicam	50 μM	Shoot induction	5.27 ± 0.45
CNQX	50 μM	Seed germination	6.12 ± 0.38
CNQX	50 μM	Co-cultivation	6.08 ± 0.68
CNQX	50 μM	Shoot induction	6.05 ± 0.30
LaCl_3_	10 mM	Seed germination	6.14 ± 0.43
LaCl_3_	10 mM	Co-cultivation	6.06 ± 0.33
LaCl_3_	10 mM	Shoot induction	4.77 ± 0.30 *
Trichostatin A	1 μM	Seed germination	6.10 ± 0.63
Trichostatin A	1 μM	Co-cultivation	5.98 ± 0.33
Trichostatin A	1 μM	Shoot induction	5.95 ± 0.45

Values are presented as mean ± SD (n = 3). Asterisk (*) indicates a significant difference compared to the CK according to a two-tailed Student’s *t*-test at *p* < 0.05.

**Table 3 plants-14-03802-t003:** Mean number of regenerated shoots per explant under different 6-BA and NAA concentrations with or without *Agrobacterium* infection.

6-BA (mg/L)	NAA (mg/L)	Non-Infection	Infection
4	0.0	0.22 ± 0.24 ^e^	0.14 ± 0.21 ^e^
4	0.2	12.05 ± 0.73 ^a^	5.82 ± 0.64 ^c^
4	0.4	9.12 ± 0.64 ^b^	6.63 ± 0.55 ^b^
4	0.5	6.45 ± 0.49 ^c^	7.43 ± 0.49 ^a^
4	0.6	3.15 ± 0.41 ^d^	5.88 ± 0.38 ^c^
2	0.2	8.76 ± 0.58 ^b^	4.20 ± 0.52 ^d^
6	0.2	9.15 ± 0.57 ^b^	4.55 ± 0.55 ^d^

Values are presented as mean ± SD (n = 3). For each column, different letters after SD indicate significant differences among treatments according to LSD test at the 0.05 level.

**Table 4 plants-14-03802-t004:** Transformation efficiency using selectable markers and visual reporters.

Selection Method	Vector	Regenerated Shoots per Explant	Non-Transformed Shoots/ Regenerated Shoots (%)	Transformation Efficiency (%)
CK (no infection, no selection)	NA	12.03 ± 0.62 ^a^	NA	NA
Kanamycin	pCAMBIA2300-eYGFPuv	0.45 ± 0.14 ^c^	82.81 ± 12.31 ^b^	6.67 ± 3.06 ^b^
Hygromycin	pCAMBIA1300-eYGFPuv	0.19 ± 0.03 ^c^	84.72 ± 16.67 ^b^	2.67 ± 3.06 ^b^
eYGFPuv	pCAMBIA2300-eYGFPuv	8.55 ± 0.72 ^b^	97.72 ± 0.56 ^a^	19.33 ± 4.16 ^a^
RUBY	pCAMBIA2300-RUBY	9.77 ± 0.44 ^b^	97.95 ± 0.46 ^a^	20.00 ± 4.62 ^a^

Values are presented as mean ± SD (n = 3). For each column, different letters after SD indicate significant differences among treatments according to LSD test at the 0.05 level.

**Table 5 plants-14-03802-t005:** Agronomic trait measurement of transgenic lines.

Line	Growth Period (Days)	Gross Weight (kg)	Net Weight (kg)	Head Height (cm)	Head Width (cm)	Number of Head Leaves
L68	61.40 ± 6.22 ^a^	1.98 ± 0.13 ^a^	1.49 ± 0.08 ^a^	25.51 ± 1.66 ^a^	13.43 ± 0.83 ^a^	27.33 ± 1.08 ^a^
RB16	53.87 ± 3.15 ^b^	1.32 ± 0.11 ^c^	0.92 ± 0.11 ^c^	16.60 ± 0.90 ^c^	8.91 ± 0.39 ^c^	24.73 ± 0.60 ^a^
RB25	54.53 ± 2.01 ^b^	1.72 ± 0.11 ^b^	1.32 ± 0.13 ^b^	22.79 ± 1.04 ^b^	12.45 ± 0.18 ^b^	25.00 ± 0.90 ^a^
RB52	47.93 ± 1.12 ^c^	0.91 ± 0.11 ^d^	0.72 ± 0.12 ^c^	11.87 ± 0.51 ^d^	6.65 ± 0.19 ^d^	26.80 ± 0.69 ^a^

Values are presented as mean ± SD (n = 3). For each column, different letters after SD indicate significant differences among treatments according to LSD test at the 0.05 level.

## Data Availability

The original contributions presented in this study are included in the article/Appendix A. Further inquiries can be directed to the corresponding author.

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
