# Peer review of "Enhanced Agrobacterium-Mediated Transformation in Chinese Cabbage via Tenoxicam, Phytohormone Optimization, and Visual Reporters"

_plants, 2025, doi:10.3390/plants14243802_

Round 1

Reviewer 1 Report

Comments and Suggestions for Authors

Chinese cabbage's recalcitrance to Agrobacterium-mediated transformation hinders functional genomics. This study identified infection and antibiotic selection as major bottlenecks, severely inhibiting cotyledonary petiole regeneration. This integrated protocol achieved high transformation efficiency (19.33-20.00% vs. 2.67–6.67% with antibiotics). Stable RUBY-overexpressing lines were generated, showing biomass inversely related to expression. The transgenic modification of Chinese cabbage has always been a difficult problem to solve. This research is highly innovative and provides new references and thoughts for future genetic transformation of Chinese cabbage. There are still a few minor issues that need to be corrected.
1.Is it the expression "salicylic-acid-signalling inhibitor tenoxicam" in the abstract accurate? The original text clearly states in line 61 that tenoxicam is a "JA signaling inhibitor" (jasmonic acid signaling inhibitor), which creates a contradiction.
2.Format writing issues: line 194 Figue 2; line 139, the comma following 'selection'; line 148, the comma following 'evaluation'; there are multiple grammatical errors (for example, "3. Result" should be "3. Results"; "Howerver" should be "However"; "ennable" should be "enable"); and the indentation of the first line in some paragraphs is inconsistent; Line 129: "reangents" is changed to "reagents"
3.Section 4.3 merely states that "tenoxicam enhances regeneration when applied during seed germination", but does not explain why it is ineffective in other stages (such as co-culture/induction)
4.The terms "genetic transformation efficiency" and "transformation frequency" are used interchangeably throughout the text (e.g. in lines 21-22 of the abstract vs. the title of Table 4), and they need to be standardized.
5.Line 162, the significance level of the LSD test is not clearly specified in several places. It can be supplemented as "The significance level of the LSD test is set to P < 0.05".

Author Response

Please see the attachment." in the box if you only upload an attachment. 

Reviewer 2 Report

Comments and Suggestions for Authors

The manuscript provides essential information on the genetic transformation of Chinese cabbage. However, the document requires substantial language and scientific improvements.

Eg: in vitro should be italic

Abstract: Need to improve both English and scientific writing

Figure 1D: Need quality improvement

Author Response

(The authors gave the same response as above.)

Reviewer 3 Report

Comments and Suggestions for Authors

In this paper, an efficient Agrobacterium-mediated genetic transformation method for Chinese cabbage was developed, which significantly improved the transformation efficiency by introducing tenoxicam, optimizing plant hormone concentrations, and using visual reporter genes eYGFPuv and RUBY, providing an important tool for functional genomics research and gene editing of Chinese cabbage. Chinese cabbage is considered one of the most difficult species in the genus Brassica to undergo Agrobacterium-mediated transformation. Over the past three decades, although conversion efficiency has been slightly improved in some genotypes through genotypic screening, media optimization, and protocol improvement, there has been a significant increase in conversion efficiency in some genotypes, but it has been significantly improved compared to rapeseed (B. napus) and cabbage (B. oleracea), the genetic transformation of Chinese cabbage still faces great challenges. In addition, the antibiotic selection system has problems such as high false positive rate and inhibitory effect on the growth and differentiation of transformed cells, while the use of developmental regulators may lead to pleiotropic effects (such as decreased fertility, leaf curling, etc.), further limiting its application. The study used 110 Chinese cabbage germplasm resources, including 29 commercial varieties and 71 inbred lines. The regenerative ability of copetioles was evaluated by in vitro culture, and it was found that there were significant differences in regeneration efficiency between different genotypes. The regenerative inhibitory effects of three Agrobacterium strains (GV3101, LBA4404, and K599) on genotypes L68, L69, and L131 with high regenerative capacity were tested. The effects of a variety of plant immune signaling inhibitors (such as tenoxicam, CNQX, LaCl3, and trichostatin A) on cotylard regeneration after Agrobacterium infection were evaluated. Comparison of genetic transformation efficiency The study compared the performance of antibiotic selection markers (kanamycin and hygromycin) with visual reporter genes (eYGFPuv and RUBY) in genetic transformation. Genetic isolation and phenotypic analysis were performed on three representative T0 generation RUBY overexpressing plants (RB16, RB25 and RB52) and their offspring. Several agronomic traits of T2 generation transgenic plants were measured, including growth period, gross weight, net weight, leaf height, leaf width and leaf number. It was found that the pleiotropic effect caused by overexpression of RUBY could be mitigated by regulating the expression mode of RUBY. In this study, an efficient genetic transformation system of Chinese cabbage was successfully established by combining tenoxicam treatment, NAA concentration optimization and visual reporter gene screening. This system significantly improves the regenerative ability and transformation efficiency of explants, laying the foundation for functional genomics research and the application of gene editing technology. As a visual reporter gene, RUBY can not only improve transformation efficiency, but also be used to create new red germplasm resources.

Minor parts:

Line 182, figure 1, figure 1d was drawn by excel software, please redraw by Originpro, R or Sigmaplot software.

Table 1,2,3,4,5 should be use R software for the Least Significant Difference (LSD) multiple-comparison test. Please remember Lowercase letters are set as superscripts.

Author Response

(The authors gave the same response as above.)

Reviewer 4 Report

Comments and Suggestions for Authors

The work addresses a long-standing bottleneck, recalcitrant transformation of Brassica rapa ssp. Pekinensis, and provides an immediately applicable protocol that lifts efficiency from ~5% to ~20%. The combination of immune-modulator (tenoxicam), adjusted NAA, and antibiotic-free visual selection is novel for this species and will be of broad interest to the Brassica community. The work is carefully executed and the protocol will be very useful. Statistical design is basically sound, repeats are adequate and the raw data have been supplied. I recommend MINOR REVISION for this manuscript.

Major Strengths

  1. The discovery that tenoxicam, when applied during seed germination, significantly enhances post-infection regeneration (37.90% increase) and Agrobacterium infection efficiency is a key contribution. This timing-specific effect adds new insights to the field of plant transformation.
  2. The observation that the optimal NAA concentration shifts from 0.2 mg/L (non-infected) to 0.5 mg/L (infected explants) fills a critical knowledge gap, as hormone balance disruption by Agrobacterium is often overlooked.
  3. Replacing antibiotics with eYGFPuv/RUBY reduces phytotoxicity and boosts transformation efficiency to 19.33–20.00% (vs. 2.67–6.67% with antibiotics), demonstrating a more efficient and user-friendly screening method.
  4. The generation of stable RUBY-overexpressing lines (e.g., RB25) with minimal yield penalty (due to inner leaf-specific expression) provides valuable germplasm for future breeding.

Major Comments

  1. Abstract claims "reduced false-positive shoots" but visual reporters (eYGFPuv/RUBY) have higher false-positive rates (97.72–97.95%) than antibiotics.
  2. In section 3.2, A. rhizogenes K599 caused less regeneration inhibition (30.98–50.17% reduction) than A. tumefaciens strains but was excluded from subsequent experiments. Explain K599’s exclusion or test its combination with tenoxicam to explore synergies.
  3. In section 3.3, the manuscript notes tenoxicam only works during seed germination but does not explain why. Discuss potential mechanisms.
  4. In section 3.4, kanamycin and hygromycin were used at a single concentration each. Please cite the selection conditions and justify them with literature.
  5. The application value of the RUBY transgenic line RB25 has been underemphasized, and please elaborate on its application value in the Discussion section.
  6. The manuscript frequently reports means with standard deviations but often lacks clear indications of biological and technical replicates. For example, in Table 1 and Table 4, it is unclear whether "n = 3" refers to biological or technical replicates. This must be explicitly stated.

Minor Comments

  1. Seed Germination Medium composition is incomplete.
  2. Agrobacterium infection time is vague ("briefly dipped"). Define as "dip cotyledonary petioles in Agrobacterium solution (OD600 = 0.2) for 5 seconds".
  3. Specify the UV torch wavelength (365 nm?) and emission filter (if any) used for eYGFPuv screening.
  4. The primers for CYP76AD1 (RUBY) are listed as forward and reverse but appear identical. This is likely a typo and must be corrected.
  5. Student’s t-test” in Table 2 legend with the exact test used (two-tailed, unequal variance?)
  6. Table 3 contains duplicate rows (6-BA=4, NAA=0.2 appears twice).
  7. Table 4 has vector spelling errors ("pCMABIA" should be "pCAMBIA").
  8. In Section 3.6, "We further used qRT-PCR to detect RUBY expression levels... (Supplementary Table S1)" incorrectly cites Supplementary Table S1 (should be S7).
  9. The manuscript is generally well-written but contains occasional typographical errors. “trichomycin A” should be “trichostatin A”; “enhenced” should be “enhanced”.

Author Response

(The authors gave the same response as above.)
